# Biosynthesis of Arabinoside from Sucrose and Nucleobase via a Novel Multi-Enzymatic Cascade

**DOI:** 10.3390/biom14091107

**Published:** 2024-09-03

**Authors:** Yuxue Liu, Erchu Yang, Xiaojing Zhang, Xiaobei Liu, Xiaoting Tang, Zhenyu Wang, Hailei Wang

**Affiliations:** Henan Engineering Research Center of Bioconversion Technology of Functional Microbes, College of Life Science, Henan Normal University, Xinxiang 453007, China; liuyuxue@htu.edu.cn (Y.L.); yangechu@stu.htu.edu.cn (E.Y.); zhangxiaojing20229@stu.htu.edu.cn (X.Z.); liuxiaobei2023@stu.htu.edu.cn (X.L.); tangxiaoting@stu.htu.edu.cn (X.T.)

**Keywords:** arabinoside, vidarabine, sucrose, multi-enzyme cascade, biocatalysis

## Abstract

Arabinoside and derived nucleoside analogs, a family of nucleoside analogs, exhibit diverse typically biological activities and are widely used as antibacterial, antiviral, anti-inflammatory, antitumor, and other drugs in clinical and preclinical trials. Although with a long and rich history in the field of medicinal chemistry, the biosynthesis of arabinoside has only been sporadically designed and studied, and it remains a challenge. Here, we constructed an in vitro multi-enzymatic cascade for the biosynthesis of arabinosides. This artificial biosystem was systematically optimized, involving an exquisite pathway design, NADP^+^ regeneration, meticulous enzyme selection, optimization of the key enzyme dosage, and the concentration of inorganic phosphate. Under the optimized conditions, we achieved 0.37 mM of vidarabine from 5 mM of sucrose and 2 mM of adenine, representing 18.7% of the theoretical yield. Furthermore, this biosystem also has the capability to produce other arabinosides, such as spongouridine, arabinofuranosylguanine, hypoxanthine arabinofuranoside, fludarabine, and 2-methoxyadenine arabinofuranoside, from sucrose, and corresponding nucleobase by introducing different nucleoside phosphorylases. Overall, our biosynthesis approach provides a pathway for the biosynthesis of arabinose-derived nucleoside analogs, offering potential applications in the pharmaceutical industry.

## 1. Introduction

Nucleoside analogs are widely recognized for their diverse biological activities and have a longstanding history in medicinal chemistry [1,2]. Amid the pandemic, nucleoside analogs also demonstrated potential biological activity against coronaviruses [3,4]. Arabinose nucleosides or arabinosides constitute a significant group of nucleosides analogs known for their clinical and preclinical trials. Nucleosides consist of a nucleobase and a sugar moiety, whereas arabinosides replace ribose with arabinose, featuring an inverted configuration of the hydroxyl group at the C2′ position (‘up’ instead of ‘down’). Arabinosides, such as vidarabine (Ara-A), cytarabine (Ara-C), spongouridine (Ara-U), nelarabine, clofarabine, and fludarabine are highly demanded in the market, serving as cornerstones in anticancer and antiviral treatments [5,6,7]. Some of these arabinosides are natural products, including Ara-A, spongothymidine, and Ara-U [2]. For example, Ara-A has been isolated from various sources, such as the Caribbean sponge Tethya crypta and *Streptomyces antibioticus* NRRL 3238 [2,8].

Currently, arabinosides are primarily synthesized through chemical methods and biotransformation. Many arabinosides are synthesized via complex multistep chemical procedures starting from the respective nucleobase and arabinose [9]. These methodologies typically involve lengthy processes, generate undesired by-products, and yield low overall efficiencies. Moreover, these methods often rely on environmentally taxing organic solvents.

In this context, the use of enzyme catalysis proves advantageous. Biocatalysis, especially biocatalytic cascades, has emerged as a robust and advanced tool in pharmaceutical manufacturing [10,11,12]. Numerous biocatalytic methodologies have been developed for the synthesis of arabinosides. Typically, these approaches utilize enzymes such as nucleoside phosphorylases (NPs) or N-deoxyribosyltransferases to catalyze transglycosylation reactions. NPs, in particular, are powerful tools employed either singly or in combination in free or immobilized states, both in vivo and in vitro [5,13,14,15,16,17,18].

In these processes, the substrate arabinoside undergoes catalysis to produce nucleobase and arabinose-1-phosphate in the presence of inorganic orthophosphate. Subsequently, a new arabinoside is formed through the reaction of arabinose-1-phosphate with an acceptor nucleobase.

For instance, Ara-C was successfully synthesized using uracil phosphorylase (UP) and purine nucleoside phosphorylase 1 (PNP1) with Ara-A and cytosine as substrates [13]. Meanwhile, another strategy emerged, focusing on the one-pot transformation of pentoses and modified nucleobase into nucleoside analogs at the expense of ATP, utilizing ribokinase (RK), phosphopentomutase (PPM), and NP [19]. These multi-enzymatic systems are highly efficient, achieving high conversion rates or yields. However, the synthesis of arabinosides via this multi-enzyme system requires a high concentration of arabinose, up to 60 mM, which is 176 times higher than the concentration of nucleobases used. Additionally, this system is limited to the use of modified bases for synthesizing arabinosides. Our experiments further revealed that natural nucleobases are difficult to incorporate into arabinose for the synthesis of arabinosides using this system. Therefore, achieving the de novo synthesis of arabinosides through those biotransformation processes remains challenging.

To closely mimic intracellular metabolism for the future de novo synthesis of arabinosides, this study proposes a novel multi-enzymatic cascade comprising sucrose phosphorylase (SP), phosphoglucomutase (PGM), glucose-6-phosphate 1-dehydrogenase (G6PD), 6-phosphogluconate dehydrogenase (6PGDH), D-arabinose 5-phosphate isomerase (API), PPM, NP, and H_2_O-forming NADPH oxidase (TPNOX). This cascade aims to produce expensive arabinose nucleosides from inexpensive sucrose and nucleobases.

The artificial biosystem was tested, analyzed, and systematically optimized. Ultimately, Ara-A was successfully synthesized from sucrose and adenine with an adenine molar utilization rate of 18.7%. Furthermore, introducing different nucleobase substrates with NP resulted in the production of various arabinosides from sucrose. Interestingly, when SP is omitted, and the system is applied intracellularly, the system can easily be adapted into a de novo biosynthesis system of arabinoside starting from glucose. Overall, our results demonstrate a promising pathway for the de novo biosynthesis of arabinose-derived nucleoside analogs, potentially benefiting the pharmaceutical industry.

## 2. Materials and Methods

### 2.1. Materials and Chemicals

PrimeSTAR HS DNA polymerase and infusion snap assembly master mix were purchased from TaKaRa (Dalian, China). Sucrose, glucose 6-phosphate, glucose 1,6-biophosphate, nucleobases (adenine, guanine, uracil, cytosine, hypoxanthine, 2-methoxyadenine, 2-fluoroadenine), vidarabine (Ara-A), arabinofuranosylguanine (Ara-G), spongouridine (Ara-U), hypoxanthine arabinofuranoside (Ara-I), cysteine, carbazole, and kanamycin were purchased from Shanghai Biotech (Shanghai, China) or shyuanye (Shanghai, China). Nicotinamide adenine dinucleotide phosphate (NADP^+^), isopropyl-β-thiogalactoside (IPTG), methylthiazolyldiphenyl-tetrazolium bromide (MTT), and phenazine ethosulfate (PES) were purchased from Sigma-Aldrich (St. Louis, MO, USA). All other chemical reagents were analytical grade and commercially available.

### 2.2. Strains and Plasmids

The bacterial strains and plasmids utilized in this study are given in Appendix A. *Escherichia coli* strain DH5α was employed for plasmid construction, while BL21(DE3) was utilized for the overexpression of recombinant proteins. 

The plasmids pET28a-*Ec*PNP, pET28a-*Ec*UP, and pET28a-*Kl*PNP, encoding PNP1 from *E. coli* (*Ec*PNP), UP from *E. coli* (*Ec*UP), and PNP from *Klebsiella* (*Kl*PNP), respectively, were sourced from our laboratory collection. Codon-optimized genes encoding sucrose phosphorylase from *Alloscardovia omnicolens* (*gtfA*), phosphopentomutase from *Bacillus cereus (deoB)*, NADPH oxidase (a mutant of H_2_O-forming NADH oxidase from *Lactobacillus brevis*, *TPNOX*) [20], purine phosphorylase from *Alicyclobacillus acidoterrestris* (*deoD*), and ribokinase isoform 1 from *Homo sapiens* (*Hs*RK) were synthesized by Shanghai Biotech (Shanghai, China) and inserted into plasmid pET28a between the *Nco I* and *Xho I* site, in-frame with the C-terminal his-tag, to generate the corresponding recombinant plasmids pET28a-*Ao*SP, pET28a-*Bc*PPM, pET28a-TPNOX, pET28a-*Aa*PNP, and pET28a-*Hs*RK, respectively. The primers listed in Appendix A were designed and employed to obtain the corresponding gene and plasmid fragments. The seamless cloning method was employed for other plasmid constructions. All genes were individually inserted into plasmid pET28a with a C-terminal his-tag for expression in *E. coli*. Subsequently, the plasmids were transferred into *E. coli* BL21(DE3) to generate the corresponding recombinant strains for protein expression. All strains were cultivated in Luria–Bertani (LB) broth (10 g/L tryptone, 5 g/L yeast 36 extract, and 10 g/L NaCl) medium at 37 °C and 180 rpm. When necessary, 50 μg/mL of kanamycin was supplemented.

### 2.3. Protein Expression and Purification

Proteins were overexpressed in *E. coli* BL21(DE3) by initially growing cells at 37 °C in LB broth supplemented with 50 μg/mL of kanamycin until an absorbance of 0.4 to 0.6 at A_600_ was reached. Subsequently, 0.5 mM of IPTG was added, and the strain was cultivated at 20 °C for 12 h to induce protein expression. Cells were washed and resuspended in a native binding buffer (50 mM NaH_2_PO_4_, 500 mM NaCl, and pH 8.0). Cell pellets were disrupted by sonication, and intact cells and cell debris were removed through centrifugation at 12,000× *g* for 40 min at 4 °C. The supernatant was applied to a Nickel column equilibrated with a native binding buffer. The target protein was purified using the Ni-NTA purification kit (Invitrogen, Carlsbad, CA, USA). Purified proteins were stored in 50 mM of Tris-Cl (pH 7.5) with 20% glycerol at −80 °C. The concentrations of purified proteins were determined using the Bradford protein assay kit (Bio-Rad, Hercules, CA, USA), with bovine serum albumin as the standard. The purity was analyzed by SDS-PAGE.

### 2.4. Enzymatic Activity Assays

The activities of PNPs for the phosphorylation of Ara-A were determined in 50 mM of the potassium phosphate buffer at pH 7.5, containing 1 mM Ara-A and 0.1 mg/mL PNP. The reaction was initiated by the addition of an enzyme solution and incubation at 40 °C for 15 min before being quenched by heating. Adenine was quantified by HPLC. The unit of PNP activity for the phosphorylation of Ara-A is defined as the amount of enzyme that released 1 μmol of guanine per minute.

The activities of PNPs for the synthesis of Ara-A were determined as follows: Firstly, D-arabinose 1-phosphate was accumulated in the reaction mixture containing 2 mM Ara-U and 1 mg/mL *Ec*UP in a 50 mM potassium phosphate buffer at pH 7.5. The reaction mixtures were incubated at 40 °C for 15 min and quenched by heating. Protein was removed through centrifugation at 12,000× *g* for 2 min at 4 °C. Secondly, a 100 μL aliquot was transferred to 100 μL of substrate solution containing 2 mM adenine and 0.4 mg/mL PNP. The reaction was incubated at 45 °C for 15 min and quenched by heating. The Ara-A production was determined by HPLC, as described below. The unit of PNP activity for the synthesis of Ara-A was defined as the amount of enzyme that produced 1 μmol of Ara-A per minute.

The activity of PPM from *Bacillus cereus* (*Bc*PPM) was defined by coupling PNP to catalyze the synthesis as follows: Firstly, D-arabinose 5-phosphate was biosynthesized in the reaction mixture (0.25 mL, 50 mM Tris-HCl, pH 8.0) containing 4 mM ATP, 5 mM D-arabinose, 0.6 mM MgCl_2_, 6 mM KCl, and 0.25 mg *Hs*RK. The reaction mixtures were incubated at 45 °C for 20 min and quenched by heating. Protein was removed through centrifugation at 12,000× *g* for 2 min at 4 °C. Secondly, a 100 μL aliquot was transferred to 100 μL of the substrate solution containing 2 mM adenine, 1.2 mM MnCl_2_, 1 mM glucose-1,6-biophosphate, 0.2 mg *Kl*PNP and 0.2 mg *Bc*PPM. The reaction was incubated at 45 °C for 45 min and quenched by heating. Ara-A production was determined by HPLC, as described below. The unit of *Bc*PPM activity was defined as the amount of enzyme that produced 1 μmol of Ara-A per minute.

### 2.5. Production of Arabinosides from Sucrose or G6P

To verify the feasibility of Ara-A production from G6P, the reaction was conducted in 100 μL volumes containing 2 mM MgCl_2_, 0.02 mM NADP^+^, 2 mM G6P, 2 mM adenine, 10 μM *Ec*G6PD, 10 μM *Ec*6PGDH, 10 μM TPNOX, 10 μM *Ec*API, 3 U/L *Bc*PPM, and 15 U/L *Kl*PNP in a 50 mM Tris-HCl buffer (pH 8.0) at 40 °C for 2 h. *Bc*PPM was activated at a concentration 10-fold higher than that used in the reactions by incubation for 10 min at room temperature in a 50 Tris-HCl buffer (pH 8.0), 0.1 mM MnCl_2_ and 0.01 mM glucose-1,6-biophosphate before being added to the reaction mixture. Ara-A production from sucrose was performed as follows: the reaction mixture contained 2 mM MgCl_2_, 0.02 mM NADP^+^, 5 mM sucrose, 2 mM adenine, 10 μM *Ao*SP, 10 μM *Ec*PGM, 10 μM *Ec*G6PD, 10 μM *Ec*6PGDH, 10 μM TPNOX, 10 μM *Ec*API, 3 U/L of activated *Bc*PPM and 15 U/L *Kl*PNP in a 50 mM Tris-HCl buffer (pH 8.0). The reaction was initiated by the addition of 0.05 mM K_2_HPO_4_ or G6P and incubated at 40 °C for 2 h. When studying the effect of single-enzyme concentrates on Ara-A production from sucrose, the concentration of *Ec*API, *Bc*PPM, or *Kl*PNP in the above systems was adjusted. The concentration range of *Ec*API was 5 μM, 10 μM, 15 μM and 20 μM. The loading amounts of *Bc*PPM were 1.5 U/L, 3 U/L, 4.5 U/L, and 6 U/L. The loading amounts of *Kl*PNP were 7.5 U/L, 15 U/L, 22.5 U/L, and 30 U/L. The effect of phosphate concentrates on Ara-A production from sucrose was examined in reaction mixtures containing 20 μM of activated *Bc*PPM with other conditions unchanged. The concentration of K_2_HPO_4_ ranged from 0.05 mM to 2 mM. The conversion rate was calculated using Equation (1).
(1)$$\mathrm{Conversion}\;\mathrm{rate}\;(\%)=\mathrm{arabinoside}\;(\mathrm{area})/\left[\mathrm{arabinoside}\left(\mathrm{area}\right)+\mathrm{residual}\;\mathrm{nucleobase}\;\left(\mathrm{area}\right)\right]\;\times \;100\%$$

### 2.6. Effect of NADP+ Concentrates on Ru5P Accumulation

To analyze the effect of NADP^+^ concentrates on Ru5P accumulation from G6P, the reaction was conducted in 200 μL volumes containing 2 mM G6P, 10 μM *Ec*G6PD, and 10 μM *Ec*6PGDH in a 50 mM Tris-HCl buffer (pH 8.0) at 40 °C. The reaction was initiated by the addition of NADP^+^ with concentrations of 0.2 mM, 1 mM, 2 mM, or 4 mM. When coupled with NADP^+^ regeneration, 10 μM TPNOX was added to the reaction mixtures, and the concentration of NADP^+^ was adjusted to 0.01 mM, 0.02 mM, 0.05 mM, or 0.1 mM.

### 2.7. Effect of Phosphate Concentrations on the Accumulation of Intermediate Metabolite

To analyze the effect of phosphate concentrations on the accumulation of NADPH with the 6-phosphogluconate production from sucrose, the reaction mixture contained 10 μM *Ao*SP, 10 μM *Ec*PGM, 10 μM *Ec*G6PD, 0.5 mM sucrose, 2 mM MgCl_2_, 0.02 mM NADP^+^, 0.4 mM MTT, and 1 mM PES. After adding phosphate, the reactions were incubated at room temperature for 1 h. NADPH was determined by monitoring the absorbance at 570 nm. Each assay was conducted at least three times.

To analyze the effect of phosphate concentrations on the Ru5P production from sucrose. The reaction mixture contained 10 μM *Ao*SP, 10 μM *Ec*PGM, 10 μM *Ec*G6PD, 5 μM *Ec*6PGDH, 0.5 mM sucrose, 2 mM MgCl_2_, and 2 mM NADP^+^. After adding phosphate, the reactions were incubated at room temperature for 1 h.

### 2.8. Analytical Methods

Nucleobases and arabinosides were quantified at 254 nm using an HPLC system equipped with an SPD 20A DAD detector (Shimadzu, Kyoto, Japan). A reversed-phase C18 column (250 mm × 4.6 mm, 5 μm) was employed, and elution was performed with a methanol/H_2_O mixture (15:85, *v*/*v*) at a flow rate of 1.0 mL/min. 

Ru5P was assessed via a 96-well microplate adaptation of the cysteine–carbazole colorimetric assay [21]. A 36 μL aliquot of quenched reaction samples was transferred to a flat-bottom assay plate containing 100 μL of a freshly prepared H_2_SO_4_–cysteine–carbazole mixture (25 N H_2_SO_4_: 1.5% (*w*/*v*) aqueous cysteine solution: 0.12% (*w*/*v*) ethanolic carbazole solution = 23:1:1) in each well. Plates were thoroughly mixed, and the appearance or disappearance of ketose was monitored at 540 nm after 1 h of color development with heating. All assays were performed at least three times.

## 3. Results and Discussion

### 3.1. Design of the Multi-Enzymatic Cascade for Arabinoside Production from Sucrose and Nucleobase

In accordance with the bacterial nucleoside salvage pathway, arabinoside can undergo conversion into a nucleobase and arabinose 5-phosphate (A5P) catalyzed by NP and PPM (Appendix A). In biocatalysis, A5P can be acquired by phosphorylating arabinose using RK, albeit at the expense of ATP and with low catalytic efficiency [22]. Alternatively, in bacterial metabolism, A5P serves as an intermediate metabolite in the biosynthesis of nucleotide sugars generated from ribulose 5-phosphate (Ru5P) via isomerization catalyzed by API with minimal energy consumption (Appendix A). This process was intricately linked with the oxidation stage of the pentose phosphate pathway (PPP) through Ru5P.

Anticipating the potential for the in vivo biosynthesis of arabinosides, we devised an in vitro biosynthetic system (Figure 1) based on these natural pathways and the bioretrosynthetic process to convert sucrose and nucleobase into arabinoside [23]. Initially, we combined G6PD, 6PGDH, PPM, and NP and incorporated API to couple the oxidation stages of the nucleoside salvage pathway and the PPP pathway for arabinoside synthesis from glucose 6-phosphate (G6P) and nucleobase. However, this process led to inorganic phosphate accumulation, leading to the inhibition of PPM activity. To mitigate this contradiction, we integrated SP to convert inorganic phosphate and sucrose into glucose 1-phosphate (G1P) while concurrently employing PGM to convert G1P into G6P, ensuring continuous arabinoside production while recycling phosphate. 

Furthermore, the oxidation stage of the PPP pathway catalyzed by G6PD and 6PGDH is NADP^+^-dependent. The consumption of NADP^+^ and accumulation of NADPH hinder Ru5P accumulation, affecting final arabinoside production. Cofactor regeneration is crucial for cofactor-dependent biocatalytic reactions. While various oxidoreductases, such as Baeyer–Villiger monooxygenases and alcohol dehydrogenase, are employed for NADP(H)regeneration, they often introduce additional substrates and products, increasing system complexity [24]. To address this complexity, we introduced H_2_O-forming NADPH oxidase TPNOX into the system for NADP^+^ regeneration [20]. Ultimately, we assembled an eight-enzyme biosynthetic pathway encompassing inorganic phosphate and NADP^+^ regeneration cycles for arabinoside production from sucrose and nucleobase. This integrated approach not only enhances arabinoside synthesis efficiency but also minimizes byproduct formation and simplifies cofactor regeneration, making it a promising strategy for industrial-scale production.

### 3.2. Enzyme Selection of NP

The substrate spectra of NPs and their potential in the production of pharmaceutically active compounds have been extensively studied [15,25]. As demonstrated by several examples in the literature, purine nucleoside phosphorylases (PNPs) sourced from *Aeromonas hydrophila*, *Thermus thermophilus HB27*, and *E. coli*, as well as uridine phosphorylase (UP) from *Clostridium perfringens*, have been utilized in the synthesis of arabinosides or as base-modified arabinosides [16,19,22].

In our previous unpublished research, we evaluated the hydrolytic capabilities of eight NPs toward arabinosides. These NPs included PNP from *Klebsiella* (*Kl*PNP), PNPs from *E. coli* (*Ec*PNP and *Ec*XP), UP from *E. coli* (*Ec*UP), PNP from *Alicyclobacillus acidoterrestris* (*Aa*PNP), TP from *E. coli* (*Ec*TP), PNP from *Bacillus subtilis* (*Bs*PNP), and PNP from *Bacillus licheniformis* (*Bl*PNP). These NPs demonstrate varying substrate specificities for arabinosides (Table 1), offering valuable insights into arabinoside production. Notably, *Aa*PNP, *Kl*PNP, *Bs*PNP, and *Bl*PNP exhibited hydrolyzing activity towards Ara-A, Ara-G, and Ara-I. Subsequently, *Aa*PNP, *Kl*PNP, *Bs*PNP, and *Bl*PNP were further tested for their hydrolysis and synthesis activities towards Ara-A in a phosphate-buffer system (Figure 2A). Their hydrolytic activities towards Ara-A were high, ranging from 262 to 387 U/mg, whereas their synthetic activities were relatively low, ranging from 60 to 102 U/mg. The synthetic activity represented only about 25% of the hydrolytic activity, indicating unfavorable conditions for the biosynthesis of Ara-A. For the subsequent proof-of-concept, *Kl*PNP was selected for catalyzing the synthesis of Ara-A.

### 3.3. Feasibility Verification of Ara-A Production from Sucrose and Adenine

G6PD from *E. coli* (*Ec*G6PD), 6PGDH from *E. coli* (*Ec*6PGDH), API from *E. coli* (*Ec*API), PPM from *Bacillus cereus* (*Bc*PPM), *Kl*PNP, TPNOX, SP from *Alloscardovia omnicolens* (*Ao*SP), and PGM from *E. coli* (*Ec*PGM) were overexpressed in *E. coli* BL21(DE3) and purified to homogeneity (Table 2 and Figure 2B). Initially, *Ec*G6PD, *Ec*6PGDH, *Ec*API, *Bc*PPM, *Kl*PNP, and TPNOX were combined to investigate the feasibility of Ara-A production from G6P and adenine. Subsequently, all eight enzymes were combined to assess Ara-A production from sucrose and adenine. Considering the optimal activity conditions for most enzymes, reactions were conducted in a 50 mM Tris-HCl buffer (pH 8.0) containing enzymes, 20 μM NADP^+^, 2 mM MgCl_2_, 0.1 mM MnCl_2_, 2 mM adenine and 2 mM G6P or sucrose at 40 °C. Each enzyme was used at a concentration of 10 μM. When sucrose served as the substrate, the reaction was initiated by adding 0.05 mM potassium phosphate or G6P. After a 2 h reaction, Ara-A was successfully generated from adenine and G6P or sucrose (Figure 3). The results of this proof-of-concept experiment indicate the viability of our proposed enzymatic cascade for Ara-A production from sucrose and adenine.

### 3.4. Optimization of Ara-A Production from Sucrose and Adenine

In our eight-enzyme catalytic system, the NADP^+^ regeneration cycle serves not only to drive preceding reactions but also to facilitate the accumulation of Ru5P. Initially, we studied the effect of NADP^+^ concentrates on the accumulation of the intermediate metabolite. As illustrated in Figure 4A, the accumulation of Ru5P from G6P showed a positive correlation with the concentration of NADP^+^. By coupling TPNOX for NADP^+^ regeneration, the greater accumulation of Ru5P was achieved with lower concentrations of NADP^+^ (Figure 4B). The optimal NADP^+^ concentration was 0.02 mM. 

The conversion steps from Ru5P to nucleoside arabinose and the phosphorylation of sucrose are likely to be rate-limiting in our multi-enzyme catalytic system. The conversion of Ru5P to Ara-A was mediated by three enzymes: *Ec*API, *Bc*PPM, and *Kl*PNP. The equilibrium constant for the vitro isomerization between Ru5P and A5P catalyzed by *Ec*API was determined to be 0.50 ± 0.06 [21]. Additionally, native *E. coli* PPM was thermodynamically favored by 5-phosphate, and PNP also favored nucleoside synthesis thermodynamically [29,30]. Therefore, the concentration of *Ec*API, *Bc*PPM, and *Kl*PNP in the multi-enzyme-catalyzed biosystem was optimized. Among the three enzymes, increasing the concentration of *Bc*PPM proved the most effective in enhancing overall catalytic efficiency in the in vitro multi-enzymatic cascade (Figure 4C). The production of Ara-A increased with the escalating *Bc*PPM concentration, rising from 77.8 μM to 157.1 μM as the concentration of *Bc*PPM increased from 1.5 U/L to 6 U/L. However, elevating the concentration of *Ec*API or *Kl*PNP alone did not positively impact Ara-A production, thus indicating that *Bc*PPM emerged as the key enzyme for Ara-A production.

On the other hand, inorganic phosphate influences three catalytic reactions: the sucrose phosphorylation hydrolysis process, PPM-catalyzed pentose phosphate isomerization, and NP-catalyzed phosphorylation reversible reaction. Increased phosphate concentrations promote the sucrose phosphorylation process, thereby boosting Ru5P accumulation (Appendix A). However, high concentrations of inorganic phosphate can inhibit PPM activity [29]. Furthermore, NPs catalyze the reversible phosphorolysis of arabinoside in the presence of inorganic phosphate; the presence of a high concentration of inorganic phosphate potentially promotes phosphorolysis reactions of arabinoside and is unfavorable for synthesis reactions. Hence, the inorganic phosphate concentration in the system is critical to the production rate of arabinoside. Then, the inorganic phosphate concentration in the multi-enzyme catalyzed biosystem was optimized. Biotransformation was conducted with potassium phosphate at six different concentrations (0.05 mM, 0.1 mM, 0.5 mM, 1 mM, 1.5 mM, and 2 mM). As shown in Figure 4D, the Ara-A yield initially increased and then decreased with increasing phosphate concentration. A concentration of 0.36 mM of Ara-A was achieved when the phosphate concentration was raised to 1 mM or 1.5 mM. The yield of Ara-A was 18% of the theoretical value.

### 3.5. Production of Ara-A from Sucrose and Adenine under Optimal Conditions

Expanding on the aforementioned results, we constructed an optimized in vitro multi-enzymatic cascade for the production of Ara-A from sucrose and adenine. The biotransformation was carried out in a 50 mM Tris-HCl buffer (pH 8.0), comprising 6 U/L *Bc*PPM and 15 U/L *Kl*PNP, six other enzymes at concentrations of 10 μM each, 20 μM NADP^+^, 2 mM MgCl_2_, 0.1 mM MnCl_2_, 2 mM adenine and 5 mM sucrose at 40 °C. The reaction was initiated by the addition of 1 mM potassium phosphate. As illustrated in Figure 5, Ara-A was generated at a concentration of 0.37 mM of Ara-A within 4 h under the optimized condition. The yield of Ara-A was 18.7% of the theoretical value. However, with prolonged reaction times, Ara-A tended to undergo hydrolysis, resulting in a decrease in Ara-A yield and an increase in free adenine within the system. 

In the transglycosylation reaction catalyzed by two different immobilized PNPs, the yield of Ara-A reached 53% or 88% [15,17]. In contrast, the yield of Ara-A in our enzymatic biosystem is relatively low, and there remains a significant amount of residual adenine. This lower yield can be attributed to three main factors: (1) the supply of A1P for Ara-A production is limited, and the highest conversion rate of A5P synthesis from Ru5P by API is only 50.6%. (2) PPM favors the equilibrium towards thermodynamically stable 5-phosphate, which is not conducive to A1P generation and, thus, hinders Ara-A production. (3) According to our activity analysis, most NP-catalyzed reactions favor the hydrolysis of arabinosides in the presence of inorganic phosphate. Even though we systematically optimized the dosages of key enzymes and the concentration of inorganic phosphate, the intrinsic catalytic properties of these enzymes and the equilibrium constants of the reactions have not fundamentally changed, leading to the relatively low yield of Ara-A and the residual of adenine in reality. Research has shifted the reaction equilibrium more favorably towards arabinoside synthesis, aiming for higher yields, as demonstrated by our laboratory.

### 3.6. Production of Other Arabinosides through In Vitro Multi-Enzymatic Cascade Equipped with Different NPs

In addition to Ara-A, other arabinosides, such as cytarabine, also possess an important medicinal or commercial value. To broaden the application of the in vitro multi-enzymatic cascade, we introduced different NPs and nucleobases into the aforementioned enzymatic cascade to produce other arabinosides. Based on the NP enzyme activity analysis mentioned above, we utilized *Kl*PNP to synthesize Ara-G and Ara-I, and *Ec*UP to synthesize Ara-U. *Kl*PNP and *Aa*PNP were selected as candidate catalysts for the synthesis of arabinoside analogs, 2-methoxyadenine arabinofuranoside, and fludarabine, respectively.

As depicted in Figure 6, the introduction of 2 mM guanine or hypoxanthine to replace adenine in the reaction system mentioned above resulted in the production of Ara-G and Ara-I from 5 mM sucrose. Substituting adenine with the analog 2-methoxyadenine led to the production of the corresponding arabinoside analog from 5 mM sucrose at a conversion rate of 20.6%. Additionally, by substituting *Kl*PNP with *Aa*PNP and *Ec*UP, fludarabine and Ara-U were produced from 2-fluoroadenine and uracil, respectively, using the in vitro multi-enzymatic cascade. Unfortunately, the yield of Ara-U was very low compared to other arabinosides, which was related to the thermodynamic tendency of UP to catalyze the hydrolysis of nucleosides [30]. To improve the yield of Ara-U, it may be necessary to optimize the phosphate concentration and other conditions for the multi-enzyme reaction of Ara-U synthesis.

## 4. Conclusions

In this study, we designed a multi-enzymatic cascade for arabinoside production. The primary advantage of this cascade lies in its utilization of inorganic phosphate recycling and NADP^+^ regeneration to convert sucrose and nucleobase into arabinoside. Following the optimization of the biosynthetic system, we achieved 0.37 mM Ara-A from sucrose and adenine, representing 18.7% of the theoretical yield. This system not only synthesized Ara-A but also had the capability to produce other arabinosides, such as Ara-G, Ara-I, Ara-U, fludarabine, and 2-methoxyadenine arabinofuranoside, from sucrose and the corresponding nucleobase by introducing different NPs.

## Figures and Tables

**Figure 1 biomolecules-14-01107-f001:**
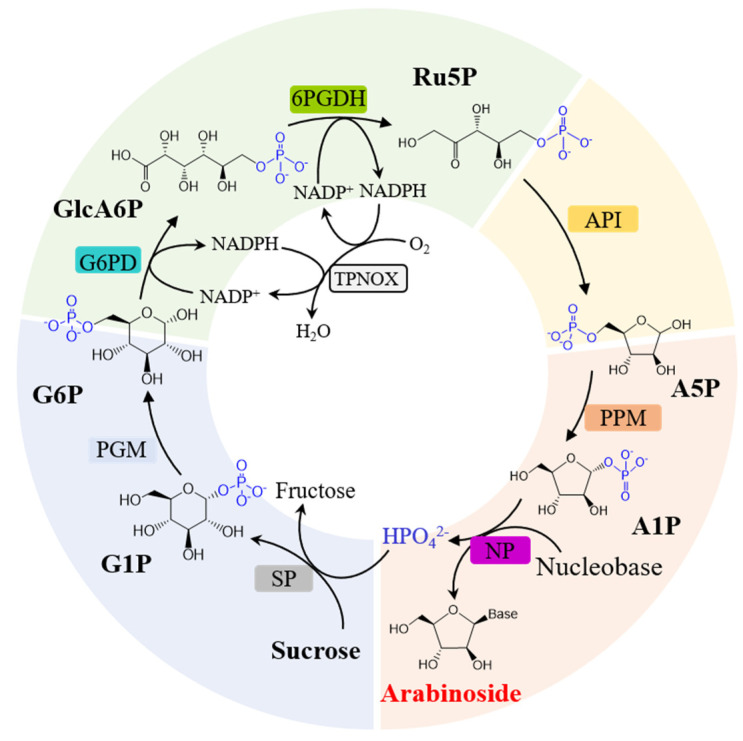
Design of the enzymatic cascade for arabinoside production from sucrose. G1P, glucose 1-phosphate; G6P, glucose 6-phosphate; GlcA6P, 6-phosphogluconate; Ru5P, ribulose 5-phosphate; A5P, arabinose 5-phosphate; and A1P, arabinose 1-phosphate.

**Figure 2 biomolecules-14-01107-f002:**
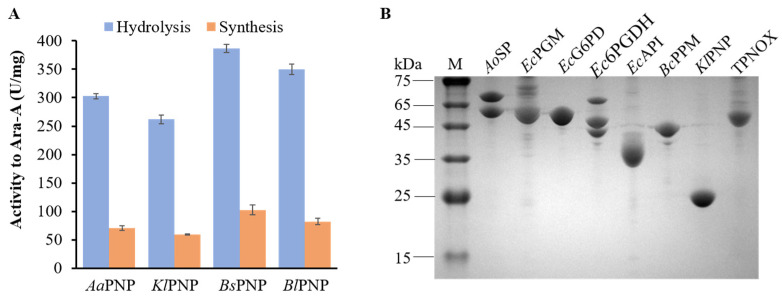
Enzymatic assays. (**A**) The activity of PNPs for the phosphorylation and dephosphorylation of Ara-A. (**B**) SDS-PAGE results in the purification of enzymes. Original images of (**B**) can be found in Appendix A.

**Figure 3 biomolecules-14-01107-f003:**
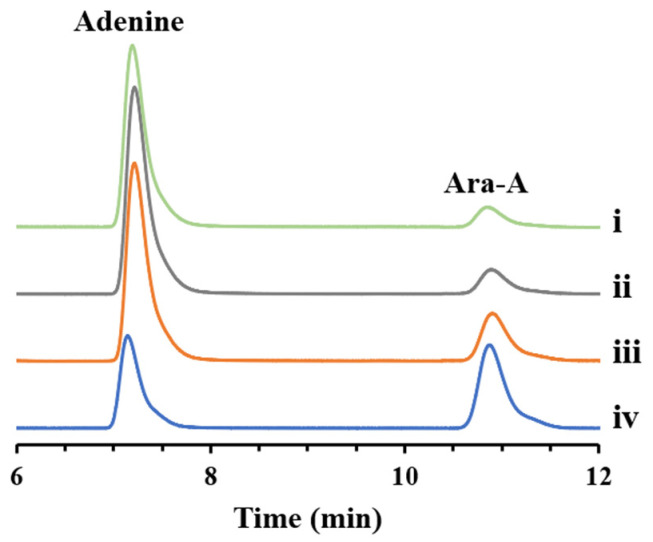
Validation of the feasibility of Ara-A production from G6P or sucrose through multi-enzyme cascade catalysis. (i), Ara-A production from G6P; (ii), Ara-A production from sucrose initiated by G6P; (iii), Ara-A production from sucrose initiated by potassium phosphate; and (iv), the authentic standard of adenosine and Ara-A.

**Figure 4 biomolecules-14-01107-f004:**
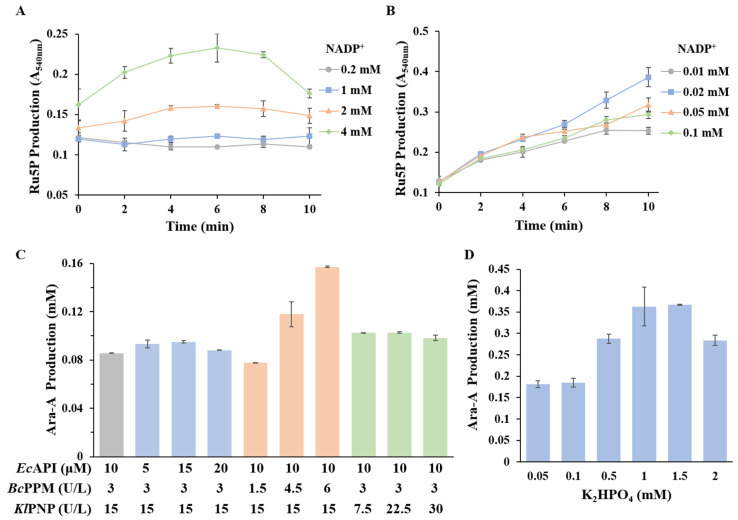
Optimization of reaction conditions for Ara-A production. (**A**) The effect of NADP^+^ concentrates on Ru5P accumulation without NADP^+^ regeneration. (**B**) The effect of NADP^+^ concentrates on Ru5P accumulation with NADP^+^ regeneration. (**C**) Effect of single-enzyme concentrates on Ara-A production. (**D**) Effect of phosphate concentrates on Ara-A production. *n* = 3 independent experiments. Data are presented as mean values ± SD.

**Figure 5 biomolecules-14-01107-f005:**
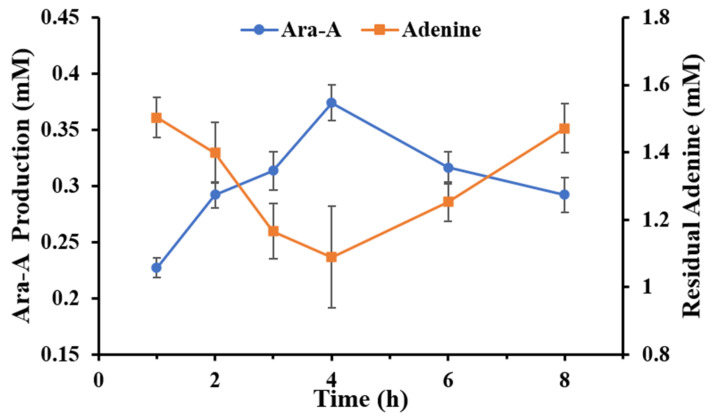
Time-course of Ara-A production using the in vitro enzymatic cascade under optimal conditions.

**Figure 6 biomolecules-14-01107-f006:**
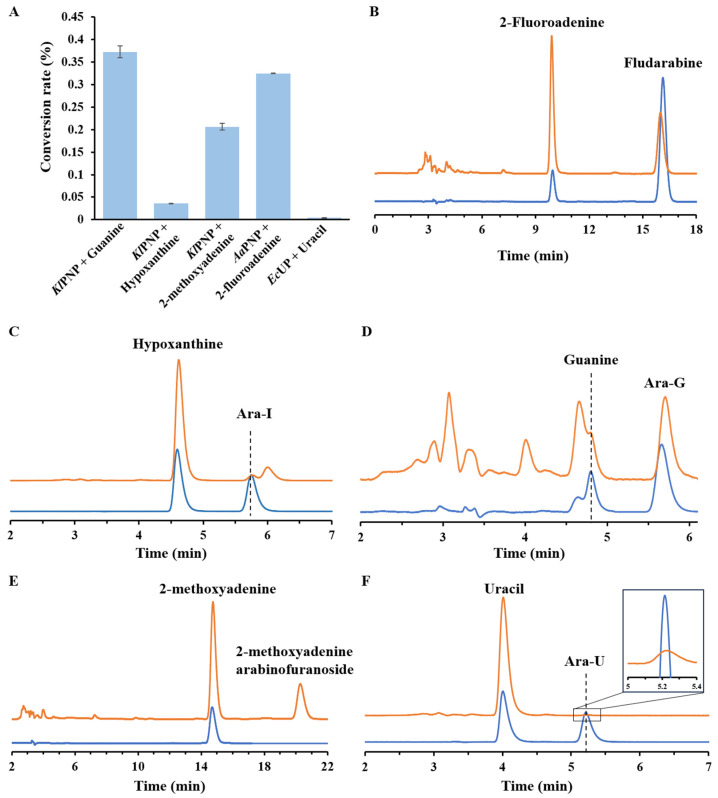
Validation of the feasibility of arabinoside production from sucrose through multi-enzyme cascade catalysis equipped with different NPs. (**A**) The conversion rate of nucleobase into arabinoside. *n* = 3 independent experiments. Data are presented as mean values ± SD. (**B**) Fludarabine production from sucrose and 2-fluoroadenine. (**C**) Ara-I production from sucrose and hypoxanthine; (**D**) Ara-G production from sucrose and guanine; (**E**) 2-methoxyadenine arabinofuranoside production from sucrose and 2-methoxyadenine; and (**F**) Ara-U production from sucrose and uracil. Blue line, the authentic standard of nucleobases and arabinosides; orange line, reaction samples.

**Table 1 biomolecules-14-01107-t001:** Substrate spectra of NPs for arabinosides.

Name	Enzyme	Source	Substrate Spectra
*Kl*PNP	Purine nucleoside phosphorylase	*Klebsiella*	Ara-A, Ara-G, and Ara-I
*Aa*PNP	Purine nucleoside phosphorylase	*Alicyclobacillus acidoterrestris*	Ara-A, Ara-G, and Ara-I
*Bs*PNP	Purine nucleoside phosphorylase	*Bacillus subtilis*	Ara-A, Ara-G, and Ara-I
*Bl*PNP	Purine nucleoside phosphorylase	*Bacillus licheniformis*	Ara-A, Ara-G, and Ara-I
*EcPNP*	Purine nucleoside phosphorylase	*E. coli*	Ara-U
*Ec*UP	Uridine phosphorylase	*E. coli*	Ara-U
*Ec* *XP*	Purine nucleoside phosphorylase 2	*E. coli*	Ara-G
*EcTP*	Thymidine phosphorylase	*E. coli*	Ara-U

**Table 2 biomolecules-14-01107-t002:** Properties of the enzymes used in the biocatalytic pathway.

Name	Enzyme	Source	Conditions Required for Activity	Reference
*Ec*PGM	Phosphoglucomutase	*E. coli*	pH 8.0, Mg^2+^	[26]
*Ec*G6PD	Glucose-6-phosphate 1-dehydrogenase	*E. coli*	/	
*Ec*6PGDH	6-phosphogluconate dehydrogenase	*E. coli*	pH 8.0, NADP^+^, Mg^2+^	[27]
*Ec*API	D-arabinose 5-phosphate isomerase	*E. coli*	pH 8.5; Zn^2+^ inhibited	[21]
*Bc*PPM	Phosphopentomutase	*Bacillus cereus*	pH 8.0, glucose 1,6-bisphosphate dependent, Mn^2+^	[28]
TPNOX	NADPH oxidase	*Lactobacillus brevis*	pH 7.5, NADPH	[20]

/ Not reported.

## Data Availability

Data will be made available on request from the corresponding author.

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
