# Peer review of "Biosynthesis of Arabinoside from Sucrose and Nucleobase via a Novel Multi-Enzymatic Cascade"

_biomolecules, 2024, doi:10.3390/biom14091107_

Round 1

Reviewer 1 Report

Comments and Suggestions for Authors

The authors of this manuscript have described an enzymatic cascade pathway for the synthesis of different arabinose nucleosides from sucrose and nucleobases. In their work, Ara-A, the nucleoside of arabinose and adenine, was was the model product for which the optimization of the enzymes and cofactor concentrations (among others) were carried out. Finally, the authors also applied their optimized conditions for different nucleobases.

The overall presentation of the work is clear and the methodologies and results are sound. The manuscript is well-written and easy to understand. However, the overall novelty of this work in comparison to the existing literature seems unclear. Here, especially literature [19] by Fateev et al. is closely related to the work presented this manuscript. As correctly stated by the authors in the introduction: "These multi-enzymatic systems are highly efficient, achieving high conversion rates or yields." (row 61).

Thus, the authors of this manuscript should explain in the introduction more clearly what the significance/ advantage of using a de novo synthesis from sucrose is (in comparison to the similar multi-enzymatic system starting from pentoses as in [19]). Is the system applied in this manuscript more flexible towards different/ natural nucleobases (in [19] only modified nucleobases were used)? Is it easier to perform? Are the materials/ enzymes cheaper? Why should one consider this method, even though the overall yield is clearly lower? These are some questions that might come to mind of readers.

If the authors can reinforce the significance of their method in the introduction, I believe the manuscript should be accepted. As I said before, the experiments performed in this work are sound and certainly of interest - the novelty is really the only question that remains to be answered in my opinion.

Author Response

Comment 1. The overall presentation of the work is clear and the methodologies and results are sound. The manuscript is well-written and easy to understand. However, the overall novelty of this work in comparison to the existing literature seems unclear. Here, especially literature [19] by Fateev et al. is closely related to the work presented this manuscript. As correctly stated by the authors in the introduction: "These multi-enzymatic systems are highly efficient, achieving high conversion rates or yields." (row 61).

Thus, the authors of this manuscript should explain in the introduction more clearly what the significance/ advantage of using a de novo synthesis from sucrose is (in comparison to the similar multi-enzymatic system starting from pentoses as in [19]). Is the system applied in this manuscript more flexible towards different/ natural nucleobases (in [19] only modified nucleobases were used)? Is it easier to perform? Are the materials/ enzymes cheaper? Why should one consider this method, even though the overall yield is clearly lower? These are some questions that might come to mind of readers.

Response 1: We appreciate your insightful comments.

      The synthesis of arabinosides from sucrose presents several significant advantages compared to the multi-enzymatic system starting from pentoses, as described in [19]. Firstly, the pentose-based system requires a very high concentration of arabinose—up to 60 mM, which is 176 times higher than the concentration of nucleobases used. Secondly, the multi-enzymatic system starting from pentoses this system is limited to using only modified nucleobases for the synthesis of arabinosides. However, our experiments have demonstrated that natural nucleobases are difficult to incorporate into arabinose for the synthesis of arabinosides using this method. Finally, our multi-enzymatic system starting from sucrose is more versatile. By removing sucrose phosphorylase (SP), the system can be adapted to use glucose as a substrate, making it more suitable for intracellular applications and expanding its potential use. Additionally, this adaptability makes the system more flexible and potentially more cost-effective due to the broader range of starting materials and enzymes.

     We have now emphasized these points more clearly in the revised manuscript to highlight the significance and advantages of our approach.

Reviewer 2 Report

Comments and Suggestions for Authors

This research presented the synthesis of arabinoside from sucrose and nucleobase using a multi-enzymatic cascade.

The reaction might be a challenge in the field, and their effort in this work is appreciated. However, after considering the contents and achievements, the reviewer suggests that it could be improved, as follows.

1. There are many statements regarding the potential for application in the pharmaceutical industry. Nonetheless, the yield obtained from the system was 0.37 mM, which was 18.7% of the theoretical yield. These numbers could be improved by process optimization or scaling up.

2. The authors could compare the achieved yield with other similar research and discuss to reinforce their achievement. 

3. The isolated yield of the key product (in this case, vidarabine) should be reported.

Author Response

Comment 1. There are many statements regarding the potential for application in the pharmaceutical industry. Nonetheless, the yield obtained from the system was 0.37 mM, which was 18.7% of the theoretical yield. These numbers could be improved by process optimization or scaling up.

Response 1: Thank you for your valuable comments. As you correctly pointed out, the yield could be improved through process optimization or scaling up, such as through enzyme optimization or immobilization. However, due to the inherently reversible catalytic properties of enzymes in our multi-enzymatic system—particularly the higher hydrolytic activity of purine nucleoside phosphorylase (PNP) compared to its synthetic activity—significant optimization would be required unless we can alter the catalytic equilibrium constant of these enzymes. To address this, our initial focus is on modifying one or two key enzymes within the system using enzyme engineering techniques. By reducing the reverse reaction rates of these enzymes, we aim to significantly increase product yields during the one-pot reaction. Once these modifications are achieved, further optimization or scaling up can be more effectively pursued. We are currently working on these enzyme modifications to improve the overall efficiency of the process

Comment 2. The authors could compare the achieved yield with other similar research and discuss to reinforce their achievement.

Response 2: We appreciate your insightful comments. We have compared the yield of vidarabine (Ara-A) obtained in our study with that reported in similar research. For instance, in transglycosylation reactions catalyzed by two different immobilized PNPs, the yield of Ara-A reached 53% or 88%. In contrast, the yield of Ara-A in our enzymatic biosystem is relatively low. This discrepancy may be due to several factors:

(1) The supply of A1P for Ara-A production is limited because the highest conversion rate of A5P synthesis from Ru5P by API is only 50.6%.

(2) Phosphopentomutase (PPM) favors the equilibrium towards the thermodynamically stable 5-phosphate, which is not conducive to A1P generation and consequently Ara-A production.

(3) As noted in our enzyme activity analysis, most nucleoside phosphorylase (NP)-catalyzed reactions favor the hydrolysis of arabinosides in the presence of inorganic phosphate.

We have included a more detailed discussion of these points in the revised manuscript (Section 3.5: Production of Ara-A from sucrose and adenine under optimal conditions).

Comment 3. The isolated yield of the key product (in this case, vidarabine) should be reported.

Response 3: Thank you for your comments. We agree that reporting the isolated yield of the key product, vidarabine, is important. However, we believe it is best to provide this information after further optimizations have been made to improve the overall yield. Additionally, there are various established methods for the separation of arabinosides, and the technology in this area is already quite advanced.

Moreover, the innovation of our multi-enzymatic system lies not in the isolated yield but in its flexibility and scalability. Our system, which starts from sucrose, can be modified to use glucose as a substrate by simply removing sucrose phosphorylase (SP). This modification makes it easier to adapt the system for intracellular applications, demonstrating its versatility.

Round 2

Reviewer 2 Report

Comments and Suggestions for Authors

The authors added a discussion regarding the concerned issue. Although the systems could be improved, this report could be a seed for further research in the relevant fields.